# GateFlow: Mitigating Shortcut Learning in VLA Models via Gated Flow Matching

## Abstract

Vision-Language-Action (VLA) models promise general-purpose robotic intelligence by leveraging pretrained vision-language representations. However, these models suffer from *shortcut learning*—exploiting spurious correlations between visual patterns and actions rather than developing semantic understanding. This occurs because VLA models optimize an Evidence Lower Bound (ELBO) proxy instead of the true likelihood, creating an optimization gap that enables memorized patterns to masquerade as genuine solutions. To mitigate this problem, we introduce **GateFlow**, a transport-guided gating mechanism that detects and suppresses shortcut learning by measuring the Wasserstein distance between observation and action representations. Low transport distance indicates semantic understanding and receives strong enhancement, while high distance reveals shortcuts and triggers suppression. This selective gating closes the ELBO-NLL gap by guiding optimization toward true likelihood minimization. We provide theoretical guarantees showing that GateFlow concentrates gradients on semantic features while eliminating spurious patterns. Empirically, GF-VLA achieves state-of-the-art performance on various tasks, with substantial improvements on long-range tasks or complex scenarios under non-stationary perturbations. GateFlow integrates seamlessly into existing VLA architectures with minimal computational overhead, offering a practical solution to more general robotic learning.

## 1 Introduction

Humans naturally integrate vision, cognition, and action when manipulating objects. When picking up a cup, we perceive its shape, recognize its fragility, and adjust our movements to different viewing angles. Recent VLA models (Brohan et al., 2023b; Kim et al., 2024; Black et al., 2024; Qu et al., 2025a; Bjorck et al., 2025) aim to replicate this capability by leveraging pretrained vision-language representations (Radford et al., 2021; Beyer et al., 2024). These models have demonstrated success across diverse tasks from kitchen cleanup to factory assembly, suggesting potential for general-purpose robotic intelligence (Bommasani et al., 2021).

However, these achievements mask a critical limitation. Recent studies reveal that VLA models trained on robot data exhibit fragile performance—minor variations in backgrounds or instruction phrasing cause catastrophic failures (Kim et al., 2024; Intelligence et al., 2025). These models succeed only within narrow training distributions, breaking immediately when conditions shift. This exposes a fundamental problem: VLA models develop *shortcut learning*, memorizing spurious correlations instead of learning genuine task understanding.

Consider the task "place the red cube in the blue container." Genuine understanding requires parsing the instruction, identifying objects by color, reasoning about spatial relationships, and planning appropriate movements. Instead, VLA models often learn shortcuts—direct mappings from pixel patterns to memorized actions that bypass semantic understanding entirely. These spurious correlations appear successful during training but fail catastrophically under distribution shift: new camera angles, different lighting, or novel object textures break these fragile associations.

The root cause extends beyond architecture or data. Modern VLA models typically employ flow matching for action generation (Lipman et al., 2023; Black et al., 2024), which learns vector fields that transform noise distributions into action distributions. However, flow matching optimizes an Evidence Lower Bound (ELBO) rather than the true negative log-likelihood (NLL) (Kingma &

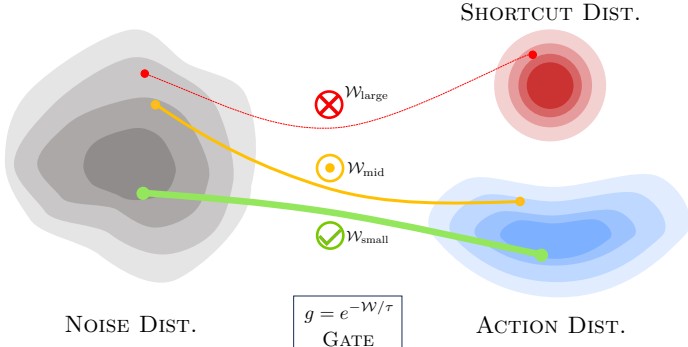

Figure 1: **GateFlow detects and suppresses shortcut learning through transport-guided gating.** Flow matching transforms noise into actions, but models can "hack" the ELBO-NLL gap by learning shortcuts to spurious correlations (red distribution) instead of genuine semantic understanding (blue distribution). GateFlow measures the Wasserstein distance $\mathcal{W}$ between observation and action features to distinguish these pathways. The gate $g = e^{-\mathcal{W}/\tau}$ selectively modulates learning: **Green path** ($\mathcal{W}_{\text{small}}$): Semantic features with low transport distance receive strong gates, enabling robust learning. **Yellow path** ($\mathcal{W}_{\text{mid}}$): Intermediate features receive moderate gating. **Red path** ($\mathcal{W}_{\text{large}}$): Shortcut features with high transport distance are suppressed, preventing spurious learning. This mitigates the ELBO-NLL gap by guiding optimization toward true likelihood minimization.

Gao, 2023; Song et al., 2021). This ELBO-NLL gap creates a fundamental vulnerability: models can "hack" the proxy objective through spurious shortcuts without developing true understanding. These shortcuts exploit specific patterns like fixed camera angles, consistent object placements, *etc.*, producing policies that succeed during training but fail catastrophically during deployment.

To address this fundamental problem, we introduce **GateFlow**, a transport-guided gating mechanism that detects and suppresses shortcut learning. Our key insight is that the alignment between observation and action representations reveals whether models employ genuine understanding or exploit spurious correlations. Semantic understanding produces smooth transformations through meaningful intermediate representations, while shortcuts create discontinuous jumps that bypass understanding entirely. We measure this alignment using optimal transport theory, specifically the Wasserstein distance, to identify and selectively modulate different learning pathways.

Specifically, GateFlow computes the Wasserstein distance $\mathcal{W}$ between observation features and action representations, quantifying the geometric effort required to transform one distribution into another. Low Wasserstein distance indicates semantic alignment, which means we get features that contain the task relationship. High distance reveals shortcuts, *i.e.*, spurious patterns that bypass understanding. This distance pattern motivates us design an exponential gating mechanism: $g = e^{-\mathcal{W}/\tau}$, which selectively amplifies semantic features while suppressing shortcuts. Figure 1 illustrates how this transport-guided gating separates genuine understanding from spurious shortcuts that exploit the ELBO-NLL gap. Our experiments validate this approach. With GateFlow, our GF-VLA model achieves state-of-the-art performance on various tasks. More importantly, our non-stationary perturbation experiments provide further evidence of shortcut suppression: 16.8% relative improvement on complex long-horizon tasks where spurious correlations cause catastrophic baseline failures.

In this work, our contributions can be mainly summarized as below:

- **Analysis of the shortcut problem:** We theoretically identify shortcut learning in VLA models as memorized patterns arising from optimizing the proxy objective, explaining why current methods produce fragile policies.
- **GateFlow method:** We introduce transport-guided gating that measures how well observations and actions align, strengthening real understanding while weakening shortcuts. Meanwhile, Gate-Flow can be integrated with minimal changes to existing models.
- **SOTA performance:** With GateFlow, our GF-VLA model shows clear improvements on various tasks, demonstrating that addressing memorized shortcuts yields more powerful VLA model.

## 2 RELATED WORK

**Vision-Language-Action Models.**  Vision Language Models (VLMs) (Zhu et al., 2023; Wang et al., 2024) typically combine LLM reasoning (Touvron et al., 2023a;b; Bai et al., 2023) with modal-specialized encoders (Radford et al., 2021; Zhai et al., 2023) for unified multimodal understanding. Pioneering works (Alayrac et al., 2022; Liu et al., 2024; 2023b; Lu et al., 2023) demonstrated strong few-shot instruction-following capability. With these visual backbones, VLA models extend them to generate robot actions from visual observations and language instructions, representing a paradigm shift in robot learning (Brohan et al., 2023c;a; Kim et al., 2024; Black et al., 2024; Driess et al., 2023). Early work like RT-1 (Brohan et al., 2023c) demonstrated transformer-based architectures could scale to diverse tasks, while RT-2 (Brohan et al., 2023a) showed web-scale vision-language pretraining improves robotic control. Recent advances follow two primary approaches: (1) *Direct action prediction*: OpenVLA (Kim et al., 2024) scales to 7B parameters with open-source datasets, Pi0 (Black et al., 2024) and its enhanced version Pi0.5 (Intelligence et al., 2025) use flow matching for continuous control, while models like Octo (Team et al., 2024) provide generalist policies across embodiments; (2) *Visual prediction*: PaLM-E (Driess et al., 2023) embeds sensors into language models, GR-2 (Cheang et al., 2024) uses video pretraining for action generation. Specialized variants address specific challenges: SpatialVLA (Qu et al., 2025b) incorporates 3D spatial representations, Pi0-Fast (Pertsch et al., 2025) improves inference efficiency through action tokenization, gr00t-N1 (Bjorck et al., 2025) targets humanoid control, and CoT-VLA (Zhao et al., 2025) adds visual chain-of-thought reasoning. Despite these advances, all approaches suffer from shortcut learning (Geirhos et al., 2020), memorizing spurious correlations instead of developing semantic understanding, particularly when trained on limited robot data.

**Flow Matching and Optimal Transport.**  Flow Matching (Lipman et al., 2023) trains neural ODEs without simulation, directly learning vector fields that transform distributions. This paradigm improves on Continuous Normalizing Flows (Chen et al., 2018; Grathwohl et al., 2018; Papamakarios et al., 2021) which require expensive ODE solving during training. Recent developments include Conditional Flow Matching (Tong et al., 2023) for simplified training with conditional distributions, Rectified Flow (Liu et al., 2022) for straighter transport paths, and Stochastic Interpolants (Albergo et al., 2023) unifying flows and diffusions. OT-CFM (Pooladian et al., 2023; Tong et al., 2024) uses optimal transport for trajectory optimization, finding better paths between distributions.

Optimal transport theory (Villani, 2009; Peyré & Cuturi, 2019) provides principled tools for comparing distributions geometrically, with computational advances making it practical: Sinkhorn distances (Cuturi, 2013) add entropic regularization while Sliced Wasserstein (Bonneel et al., 2015; Kolouri et al., 2019) projects to 1D for linear complexity. However, existing OT approaches cannot address VLA shortcut learning: OT-CFM operates at the trajectory level without addressing the ELBO-NLL optimization mismatch, while traditional methods focus on improving generation quality rather than detecting spurious feature correlations. Our key innovation uses transport distances as *alignment signals for feature gating* rather than trajectory modification, making GateFlow compatible with any flow matching variant.

## 3 PRELIMINARIES

We now establish the core concepts underlying our approach.

Vision-language models (VLMs) jointly process images and text to create shared representations through three components: a visual encoder $f_{\text{vis}} : \mathbb{R}^{H \times W \times 3} \to \mathbb{R}^{n \times d}$ that maps images to feature representations, a text encoder $f_{\text{text}} : \mathcal{V}^* \to \mathbb{R}^{m \times d}$ that transforms texts from vocabulary into feature embeddings, and cross-modal fusion $f_{\text{fuse}} : \mathbb{R}^{n \times d} \times \mathbb{R}^{m \times d} \to \mathbb{R}^D$ that integrates both modalities.

Flow Matching provides the foundation for action generation by learning to transform a simple distribution (*e.g.*, Gaussian noise) into a complex data distribution through a continuous flow. Given source distribution $p_0 = \mathcal{N}(0, \mathbf{I})$ (noise) and target distribution $p_1$ (data), flow matching learns a time-dependent vector field $v_\theta(\mathbf{x}, t)$ for $t \in [0, 1]$ that transports points from noise to data:

$$\frac{d\mathbf{x}_t}{dt} = v_\theta(\mathbf{x}_t, t), \quad \mathbf{x}_0 \sim p_0. \tag{1}$$

The training employs the following objective:

$$\mathcal{L}_{\text{CFM}} = \mathbb{E}_{t,\mathbf{x}_1,\mathbf{x}_0} \left[ \|v_\theta(\mathbf{x}_t, t) - (\mathbf{x}_1 - \mathbf{x}_0)\|^2 \right], \tag{2}$$

where $\mathbf{x}_t = (1-t)\mathbf{x}_0 + t\mathbf{x}_1$ interpolates between noise $\mathbf{x}_0 \sim p_0$ and data $\mathbf{x}_1 \sim p_1$.

Many Vision-Language-Action (VLA) models choose to combine VLMs with flow-based action generation for robot control. Given observations $\mathbf{o} = (\mathbf{o}_{\text{vis}}, \mathbf{o}_{\text{lang}}, \mathbf{o}_{\text{prop}})$ containing visual, language, and proprioceptive inputs, the VLM encoding $\mathbf{h} = f_{\text{VLM}}(\mathbf{o})$ extracts features from observations, then flow matching generates actions $\mathbf{a} \in \mathbb{R}^{d_a}$ conditioned on $\mathbf{h}$. The action flow is:

$$\frac{d\mathbf{x}_t}{dt} = v_\theta(\mathbf{x}_t, t, \mathbf{h}), \quad \mathbf{x}_0 \sim \mathcal{N}(0, \mathbf{I}). \tag{3}$$

The training minimizes:

$$\mathcal{L}_{\text{VLA}} = \mathbb{E}_{t,(\mathbf{o},\mathbf{a}),\mathbf{x}_0} \left[ \|v_\theta(\mathbf{x}_t, t, f_{\text{VLM}}(\mathbf{o})) - (\mathbf{a} - \mathbf{x}_0)\|^2 \right], \tag{4}$$

where $\mathbf{x}_t = (1-t)\mathbf{x}_0 + t\mathbf{a}$ interpolates between noise and action.

## 4  PROBLEM FORMULATION AND METHOD

### 4.1  THE ELBO-NLL GAP IN VLA MODELS

Many VLA models adopt flow matching for continuous action generation due to its superior mode coverage and training stability. However, this choice introduces a fundamental optimization challenge: flow matching optimizes an Evidence Lower Bound (ELBO) rather than the true negative log-likelihood (NLL) we actually want to minimize (Kingma & Gao, 2023; Song et al., 2021):

$$\underbrace{-\log p(\mathbf{a}|\mathbf{o})}_{\text{True NLL (want)}} \leq \underbrace{-\log p_0 - \int_0^1 \nabla \cdot v_t dt}_{\text{ELBO (optimize)}} \tag{5}$$

This gap creates a fundamental vulnerability: models can sub-optimize (or "hack") the ELBO through shortcuts that exploit spurious correlations without truly understanding the task. For instance, a VLA model might learn to associate specific lighting conditions with particular grasping motions, or correlate background objects with navigation decisions, because these patterns happen to minimize the ELBO on the training data. While such shortcuts reduce the training objective, they fail to minimize the true likelihood of correct actions given observations, leading to fragile policies that catastrophically fail when spurious patterns change. The problem is particularly critical in real robotics where training data is limited and expensive to collect.

### 4.2  TRANSPORT DISTANCE AS A SHORTCUT DETECTOR

Our key insight is that the *transport distance* between observation and action representations reveals whether a model uses semantic understanding or shortcuts.

To understand why, consider how true semantic understanding involves parsing instructions, identifying objects, and planning trajectories, creating a smooth transformation through meaningful intermediate representations. In contrast, shortcut learning directly maps pixel patterns or proprioceptive states to memorized actions, creating a discontinuous jump that bypasses understanding entirely.

The Wasserstein distance $\mathcal{W}$ precisely captures this difference by measuring the "effort" required to transform one distribution into another. When features are semantically aligned through genuine understanding, this transformation is smooth and requires little effort, resulting in low $\mathcal{W}(\mathbf{H}_{\text{obs}}, \mathbf{A})$. When features bypass understanding through shortcuts, the transformation is abrupt and costly, resulting in high $\mathcal{W}(\mathbf{H}_{\text{obs}}, \mathbf{A})$. This geometric insight, which Figure 1 visualizes through different transport paths, provides a principled way to detect and suppress shortcuts during training.

### 4.3  THE GATEFLOW SOLUTION

Based on this insight, we introduce **GateFlow**, a transport-guided gating mechanism that selectively amplifies semantic features while suppressing shortcuts. The core idea is simple: use transport

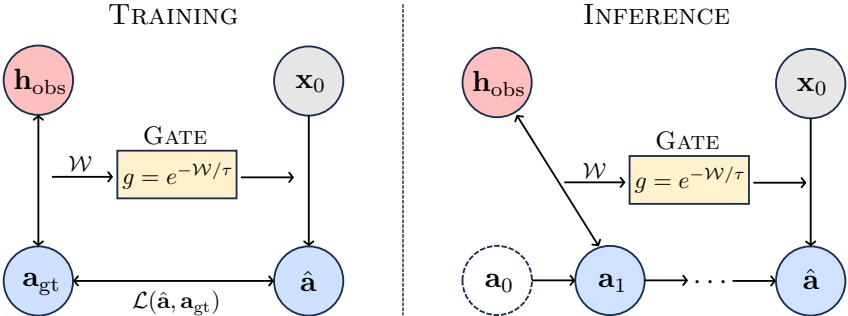

Figure 2: **GateFlow's transport-guided gating mechanism. Left (Training):** GateFlow computes the Wasserstein distance $\mathcal{W}$ between observation features $\mathbf{h}_{\mathrm{obs}}$ and ground truth actions $\mathbf{a}_{\mathrm{gt}}$ to identify semantic alignment. This distance controls the gate $g = e^{-\mathcal{W}/\tau}$, which selectively modulates flow matching from noise $\mathbf{x}_0$ to predicted action $\hat{\mathbf{a}}$. Features with low transport distance (semantic understanding) receive enhanced gradients, while high distance features (shortcuts) are suppressed. **Right (Inference):** Without ground truth, GateFlow uses iterative refinement where previous predictions $\mathbf{a}_{i-1}$ serve as pseudo-targets to compute transport distances, progressively refining actions until convergence. This maintains the same shortcut suppression during deployment.

distance to identify which features truly understand the task, then strengthen those while weakening spurious patterns.

GateFlow operates through a simple yet powerful mechanism, illustrated in Figure 2. First, we measure the alignment between observation features $\mathbf{H}$ and action features $\mathbf{A}$ using the Wasserstein distance, computed efficiently via sliced Wasserstein approximation (Kolouri et al., 2019):

$$\mathcal{W} = \frac{1}{K} \sum_{k=1}^{K} \|\mathrm{sort}(\langle \mathbf{H}, \mathbf{u}_k \rangle) - \mathrm{sort}(\langle \mathbf{A}, \mathbf{u}_k \rangle)\|^2 , \tag{6}$$

where $\mathbf{u}_k$ are random unit vectors, giving us an efficient $O(n \log n)$ approximation. This transport distance then becomes a gate value through an exponential transformation:

$$g = e^{-\mathcal{W}/\tau}, \tag{7}$$

where $\tau$ is the temperature hyperparameter to control the gate range. The elegance of this formulation is that semantically aligned features with low Wasserstein distance naturally produce gates close to $1$, enabling stronger enhancement, while shortcuts with high distance yield gates closer to $0$, effectively suppressing them.

Finally, we use these gates to selectively modulate the VLM's understanding:

$$\hat{\mathbf{H}} = \mathbf{H} + \gamma \cdot \mathcal{E}(\mathbf{H} \odot g), \tag{8}$$

where $\mathcal{E}$ is a lightweight enhancement network and $\gamma$ controls the enhancement strength.

The practical implementation differs between training and inference phases, as shown in Figure 2. During training, we leverage ground truth actions to compute transport distances, providing direct supervision for the gating mechanism. During inference, we employ iterative refinement where previous predictions serve as pseudo-targets, maintaining the same shortcut suppression principle. This elegant design integrates seamlessly into existing VLA architectures with minimal modifications, as demonstrated in Figure 3.

During inference without ground truth actions, we employ iterative refinement starting with initial actions $\mathbf{a}_0$ generated without enhancement, using each prediction $\mathbf{a}_{i-1}$ as pseudo ground truth to compute Wasserstein distances and refine to $\mathbf{a}_i$. This process converges quickly, typically requiring only less than three iterations in practice. The complete algorithmic procedures are respectively detailed in Algorithms 1 and 2 for training and inference. Hyperparameter details are provided in Appendix B.3.

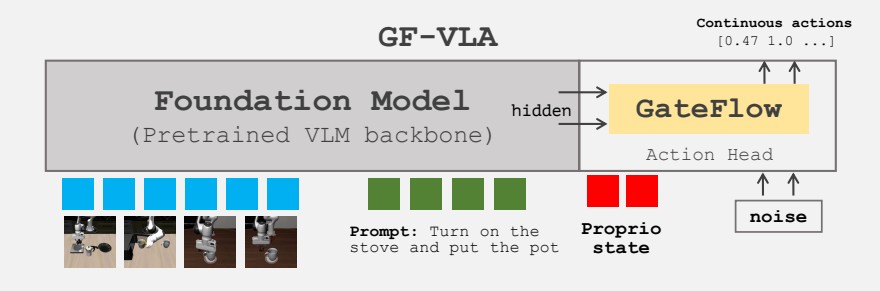

Figure 3: **GF-VLA addresses shortcut learning with minimal architectural changes.** Our Gate-Flow module integrates seamlessly between most pretrained VLM backbone and flow matching action head. The foundation model processes observation inputs into hidden representations. Gate-Flow then computes Wasserstein distances between these observation features and action targets, using transport-guided gates to suppress shortcuts while amplifying semantic understanding. This solves the fundamental ELBO-NLL optimization mismatch that enables shortcut learning.

### 4.4 WHY GATEFLOW CLOSES THE ELBO-NLL GAP

We now explain how GateFlow addresses the fundamental ELBO-NLL gap problem.

**Theorem 4.1** (GateFlow Convergence). *Flow matching with GateFlow converges to solutions that minimize both ELBO and NLL. Specifically, the gradient flow preferentially strengthens semantic features while suppressing shortcuts:*

$$\frac{\partial \mathcal{L}}{\partial \theta_{semantic}} \propto g_{high} \to 1, \quad \frac{\partial \mathcal{L}}{\partial \theta_{shortcut}} \propto g_{low} \to 0 \tag{9}$$

*This selective gradient flow closes the ELBO-NLL gap by preventing the model from exploiting spurious patterns.*

The theorem captures a fundamental reshaping of the loss landscape. Semantic features with low Wasserstein distance receive strong gradients and improve rapidly, while shortcuts with high distance receive weak gradients and cannot develop. Over time, this selective pressure guides the model toward solutions that genuinely understand the task, effectively closing the gap between what we optimize (ELBO) and what we actually want (NLL).

Three key properties make this mechanism effective. First, the sliced Wasserstein approximation reliably distinguishes shortcuts from semantic features. Second, the exponential gating creates strong differentiation between features based on their alignment quality. Third, the system exhibits self-reinforcement: as semantic features strengthen, their Wasserstein distances decrease further, creating a virtuous cycle that progressively eliminates shortcuts. Formal proofs and additional technical details are provided in Appendix A.

## 5 EXPERIMENTAL EVALUATION

We evaluate GF-VLA on robotic manipulation benchmarks to assess its effectiveness in mitigating shortcut learning and improving generalization. GF-VLA is built upon the $\pi_{0.5}$ VLA backbone model (Intelligence et al., 2025) with our proposed GateFlow module.

Our experiments mainly attempt to address three questions:

1. Does transport-guided gating improve VLA performance?

2. How does GF-VLA compare to state-of-the-art models?

3. What drives the performance gains?

Table 1: **Results on LIBERO Benchmark.** We report success rates (%) across four task suites that evaluate different generalization aspects: spatial reasoning (LIBERO-Spatial), object manipulation (LIBERO-Object), goal-oriented execution (LIBERO-Goal), and long-horizon planning (LIBERO-Long). GF-VLA achieves state-of-the-art performance by mitigating shortcut learning through transport-guided gating. All of our model's results use 50 trials per task. For baseline models, we report their best success rates from publicly available performance reports.

| Method | LIBERO-Spatial | LIBERO-Object | LIBERO-Goal | LIBERO-Long | Average |
|---|---|---|---|---|---|
| Diffusion Policy (Chi et al., 2023) | 78.5 | 87.5 | 73.5 | 64.8 | 76.1 |
| SpatialVLA (Qu et al., 2025b) | 88.2 | 89.9 | 78.6 | 55.5 | 78.1 |
| CoT-VLA (Zhao et al., 2025) | 87.5 | 91.6 | 87.6 | 69.0 | 83.9 |
| OpenVLA (Kim et al., 2024) | 84.7 | 88.4 | 79.2 | 53.7 | 76.5 |
| OpenVLA-OFT (Kim et al., 2025) | 97.6 | 98.4 | 97.9 | 94.5 | 97.1 |
| gr00t-N1 (Bjorck et al., 2025) | 94.4 | 97.6 | 93.0 | 90.6 | 93.9 |
| $\pi_0$ (Black et al., 2024) | 98.0 | 96.8 | 94.4 | 88.4 | 94.4 |
| $\pi_0$-Fast (Pertsch et al., 2025) | 96.4 | 96.8 | 88.6 | 60.2 | 85.5 |
| $\pi_{0.5}$ (Intelligence et al., 2025) | 98.8 | 98.2 | 98.0 | 92.4 | 96.9 |
| **GF-VLA (Ours)** | **99.2** | **99.0** | **98.6** | **96.4** | **98.3** |

## 5.1 EXPERIMENTAL SETUP

We conduct comprehensive experiments on LIBERO (Liu et al., 2023a), a standardized benchmark for evaluating robotic manipulation skills with a focus on generalization. LIBERO consists of four task suites (LIBERO-Spatial, LIBERO-Object, LIBERO-Goal, and LIBERO-Long) that test different aspects of model robustness, from spatial reasoning to long-horizon planning. Complete evaluation details are provided in Appendix B.2.

We compare against state-of-the-art VLA models: Diffusion Policy (Chi et al., 2023), OpenVLA (Kim et al., 2024), OpenVLA-OFT (Kim et al., 2025), SpatialVLA (Qu et al., 2025b), $\pi_0$ (Black et al., 2024), $\pi_{0.5}$ (Intelligence et al., 2025), gr00t-N1 (Bjorck et al., 2025), and CoT-VLA (Zhao et al., 2025), with $\pi_{0.5}$ as our strongest baseline.

GF-VLA is built upon $\pi_{0.5}$-base, replacing the flow matching head with GateFlow while keeping all other components identical. Our GateFlow module uses $K = 32$ sliced projections for transport distance computation and enhancement strength $\gamma = 0.4$ based on sensitivity analysis (Figure 4). We train with batch size 256 across 8 A100 GPUs for 30K steps(peak LR 5e-5, cosine decay) on the joint dataset of all four task suites. The $\pi_{0.5}$ architecture uses action horizon 10 with 7-dimensional actions for LIBERO tasks. All evaluation is based on a single checkpoint without task-specific fine-tuning. Complete hyperparameter details are in Appendix B.3.

## 5.2 MAIN RESULTS

Table 1 demonstrates that GF-VLA achieves state-of-the-art performance with 98.3% average success rate, surpassing all baselines including the strong $\pi_{0.5}$ baseline (96.9%). The results reveal important patterns about where shortcuts are most harmful. Our largest improvement occurs on LIBERO-Long (96.4% vs 92.4% for $\pi_{0.5}$), which requires extended sequential planning and is particularly vulnerable to error accumulation from spurious correlations. This 4.0% absolute improvement validates our core hypothesis that transport-guided gating helps models focus on genuine task structure.

We also observe consistent but smaller gains on perception-intensive tasks: LIBERO-Object (99.0% vs 98.2%) and LIBERO-Goal (98.6% vs 98.0%). These tasks primarily test visual understanding and instruction following rather than long-horizon planning, suggesting that while shortcuts exist in perceptual reasoning, their impact is less catastrophic than in sequential decision-making where errors compound over time.

## 5.3 ABLATION STUDIES

Table 2 validates our transport-based design through systematic ablations. Fixed gating with static values ($g = 0.5$) achieves only 92.3% average success (-6.1%), demonstrating that uniform feature modulation cannot distinguish between semantic and spurious patterns. Random gating performs

Table 2: **Comparison of Gating Mechanisms.** Success rates (%) for different gating strategies across LIBERO tasks. Transport-guided gating significantly outperforms simpler alternatives by capturing semantic alignment.

| Gating Strategy | LIBERO-Spatial | LIBERO-Object | LIBERO-Goal | LIBERO-Long | Avg |
|---|---|---|---|---|---|
| Fixed Gate ($g = 0.5$) | 96.8 -2.4% | 95.4 -3.6% | 92.2 -6.5% | 84.6 -12.2% | 92.3 -6.1% |
| Random Gate ($g \sim \mathcal{U}(0, 1)$) | 95.2 -4.0% | 93.8 -5.3% | 89.4 -9.3% | 80.8 -16.2% | 89.8 -8.7% |
| **Transport Gate (Ours)** | **99.2** | **99.0** | **98.6** | **96.4** | **98.3** |

even worse at 89.8% (-8.7%), showing that arbitrary feature suppression actively harms learning by disrupting both beneficial and harmful correlations.

Most revealing is the task-dependent degradation pattern: spatial reasoning tasks show moderate drops (LIBERO-Spatial: -2.4% to -4.0%), while complex sequential tasks suffer severe degradation (LIBERO-Long: -12.2% to -16.2%). This pattern aligns with our theoretical prediction that shortcut learning becomes increasingly detrimental as task complexity grows, confirming that effective gating requires principled measurement of observation-action alignment.

## 5.4 SHORTCUT SUPPRESSION ANALYSIS

Table 3: **Robustness to Non-Stationary Perturbations.** Success rates (%) under different time-varying noise patterns that disrupt memorized shortcuts. GateFlow shows consistent improvements across perturbation types, with most significant gains on complex long-horizon tasks.

| Perturbation | Type | LIBERO-Spatial | LIBERO-Object | LIBERO-Goal | LIBERO-Long | Avg |
|---|---|---|---|---|---|---|
| **Cosine Only** | w/o GateFlow | **87.6** | **89.8** | 86.4 | 70.2 | 83.5 |
| | w/ GateFlow | 87.4 -0.2% | 89.8 0% | **91.6** +6.0% | **80.0** +14.0% | **87.2** +4.4% |
| **Sine Only** | w/o GateFlow | 88.2 | 88.4 | 87.2 | 68.6 | 83.1 |
| | w/ GateFlow | **90.0** +2.0% | **90.6** +2.5% | **91.2** +4.6% | **80.4** +17.2% | **88.1** +6.0% |
| **Cosine + Sine** | w/o GateFlow | **87.8** | 87.2 | 86.0 | 67.8 | 82.2 |
| | w/ GateFlow | 86.4 -1.6% | **88.2** +1.1% | **90.8** +5.6% | **79.2** +16.8% | **86.2** +4.9% |

To directly evaluate GateFlow's ability to mitigate memorized shortcuts, we apply non-stationary perturbations following ([Feng et al., 2022](); [Zhang et al., 2024]()). These include cosine ($c_1 \cos(c_2 t)$), sine ($c_3 \sin(c_4 t)$), and combined trigonometric patterns with coefficients $c_i \sim \mathcal{N}(0.01, 0.5)$ applied to both visual observations and proprioceptive states. These time-varying perturbations specifically disrupt spurious correlations while preserving genuine task-relevant information.

Table 3 provides compelling evidence of shortcut suppression. GateFlow consistently improves robustness across all perturbation types, with average gains ranging from +4.4% to +6.0%. The results reveal two key insights. First, the benefits are most dramatic on long-horizon tasks (+14.0% to +17.2%), where error accumulation from shortcuts is most detrimental. Second, spatial reasoning tasks show mixed results (-0.2% to +2.0%) depending on perturbation type, suggesting that spatial shortcuts are more resilient to certain noise patterns. This confirms our hypothesis that shortcut learning compounds over extended action sequences and is most harmful in complex sequential reasoning.

## 5.5 HYPERPARAMETER SENSITIVITY ANALYSIS

Figure 4 analyzes sensitivity to GateFlow's key hyperparameters $K$ and $\gamma$, providing insights into optimal configuration and underlying mechanisms.

The sensitivity to $K$ (number of sliced projections) reveals the trade-off between computational efficiency and transport distance accuracy. Performance increases steadily from $K = 4$ to $K = 28$, then plateaus at $K = 32$. This saturation aligns with optimal transport theory: while additional projections improve sliced Wasserstein approximation quality, marginal benefits diminish as $K$ grows large.

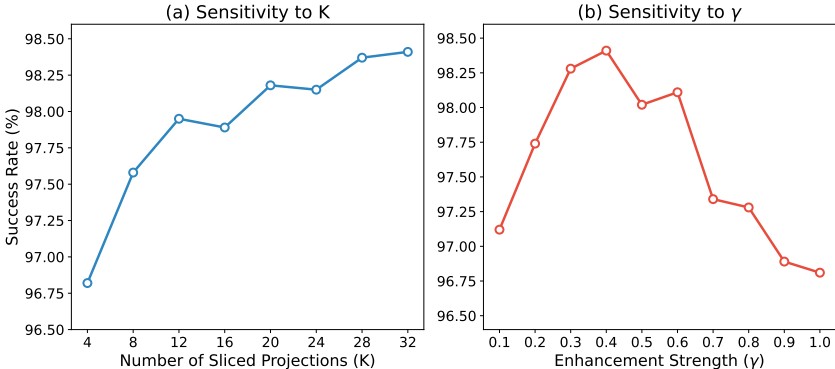

Figure 4: **Hyperparameter Sensitivity Analysis. Left:** Number of sliced projections $K$ shows increasing performance that saturates around $K = 28$, confirming that sufficient projections are needed for reliable transport distance estimation. **Right:** Enhancement strength $\gamma$ exhibits optimal performance around $\gamma = 0.4$, with degradation at higher values due to information destruction and training instability.

The $\gamma$ (enhancement strength) sensitivity exhibits more complex dynamics with a clear optimum around $\gamma = 0.4$. At low values ($\gamma = 0.1$), enhancement is too weak for effective shortcut suppression. However, the gradual decline after the peak ($\gamma = 0.7$) reveals three degradation mechanisms: **information destruction** when enhancement dominates original features, **gradient instability** from excessive magnitude, and **transport distance unreliability** as enhanced features drift from accurate approximation regions. The optimal range balances shortcut suppression with semantic information preservation.

## 6 CONCLUSION, LIMITATIONS

### 6.1 CONCLUSION

We identified a fundamental problem in Vision-Language-Action models: the optimization gap between ELBO and true likelihood enables shortcut learning, where models memorize spurious correlations instead of developing semantic understanding. Our solution, GateFlow, leverages a key geometric insight that the Wasserstein distance between observation and action representations distinguishes genuine understanding from spurious shortcuts. By using this transport distance to guide selective feature enhancement, we close the ELBO-NLL gap and force models toward semantically grounded solutions. GF-VLA validates our theoretical framework with state-of-the-art performance on LIBERO benchmarks, achieving the most dramatic improvements precisely where shortcuts are most harmful: substantial gains on complex long-horizon tasks under distributional shift. These gains come with minimal computational overhead and require no architectural changes to existing VLA models. Beyond immediate practical benefits, GateFlow demonstrates that geometric structure in representation space can guide optimization away from spurious solutions, offering a principled approach applicable to any domain where models exploit proxy objectives.

### 6.2 LIMITATIONS

While GateFlow demonstrates strong performance, several limitations warrant discussion. Our theoretical analysis assumes features have bounded norms $\|\mathbf{H}\|_2, \|\mathbf{A}\|_2 \leq R$, which holds for normalized representations but may require additional scaling mechanisms for unbounded features. The transport distance computation relies on batch-level statistics that can be influenced by outliers or small batch sizes, suggesting future exploration of running statistics or robust estimators. Additionally, GateFlow requires ground truth actions during training to compute meaningful transport distances, limiting direct extension to self-supervised or reinforcement learning settings without alternative alignment signals.

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

## A   THEORETICAL ANALYSIS

We provide detailed theoretical analysis and proofs for the main theorem. Our analysis shows how GateFlow closes the ELBO-NLL gap through three key properties: reliable shortcut detection, selective gradient flow, and convergence to NLL-optimal solutions.

### A.1   DETAILED STATEMENT OF MAIN THEOREM

We first provide the complete formal statement of Theorem 4.1 from the main text:

**Theorem A.1** (GateFlow Properties - Full Version). *Let $v_\theta$ be the velocity field learned by flow matching with GateFlow gating, where the gate $g = \exp(-\mathcal{W}/\tau)$ is computed using Wasserstein distance with $K$ random projections. Under the following conditions:*

- *Features have bounded support: $\|\mathbf{H}\|_2, \|\mathbf{A}\|_2 \leq R$*

- *The enhancement network $\mathcal{E}$ and projections $\mathcal{P}$ are Lipschitz continuous*

- *The temperature parameter satisfies $\tau > 0$*

*The following properties hold:*

*Property 1 (Concentration of Transport Estimator): The sliced Wasserstein approximation concentrates around the true Wasserstein distance with high probability:*

$$\mathbb{P}\left[|\mathcal{W} - \mathcal{W}^{true}(\mathbf{H}, \mathbf{A})| > \epsilon\right] \leq 2\exp\left(-\frac{K\epsilon^2}{8R^4}\right) \tag{10}$$

**Property 2 (Gradient Selectivity):** *The expected gradient magnitude is modulated by the gate value:*

$$\mathbb{E}[\|\nabla_\theta \mathcal{L}\|] \propto \mathbb{E}[g] = \mathbb{E}[\exp(-\mathcal{W}/\tau)] \tag{11}$$

*For semantic features with low Wasserstein distance $\mathcal{W}_{2,semantic}$ and shortcuts with high distance $\mathcal{W}_{2,shortcut}$:*

$$\frac{\mathbb{E}[\|\nabla_\theta \mathcal{L}_{semantic}\|]}{\mathbb{E}[\|\nabla_\theta \mathcal{L}_{shortcut}\|]} = \exp\left(\frac{\mathcal{W}_{2,shortcut} - \mathcal{W}_{2,semantic}}{\tau}\right) \gg 1 \tag{12}$$

**Property 3 (Convergence to NLL-Optimal Solutions):** *Under gradient flow dynamics with learning rate $\eta$, the ELBO-NLL gap decreases monotonically:*

$$\frac{d}{dt}\left(\mathcal{L}_{ELBO} - \mathcal{L}_{NLL}\right) \leq -\eta \cdot \mathbb{E}[g \cdot \|\nabla \mathcal{L}_{shortcut}\|^2] < 0 \tag{13}$$

*Asymptotically, the model converges to solutions that minimize both objectives.*

## A.2 SUPPORTING LEMMA

**Lemma A.2** (Non-expansiveness of Sorting). *For vectors $a, b \in \mathbb{R}^N$: $\|\sigma(a) - \sigma(b)\|_2 \leq \|a - b\|_2$.*

*Proof.* The sorted distance $\|\sigma(a) - \sigma(b)\|_2^2$ minimizes over permutations $\|a - Pb\|_2^2$. Taking $P = I$ gives the bound. See §2.1 of (Santambrogio, 2015). $\qquad\square$

## A.3 PROOF OF THEOREM 4.1: TRANSPORT-GUIDED SELECTIVE LEARNING

We prove each property of the main theorem separately.

### A.3.1 PROOF OF PROPERTY (I): CONCENTRATION OF TRANSPORT ESTIMATOR

*Proof.* Consider the projection-based Wasserstein distance computation with $K$ random directions $u_1, \ldots, u_K$ drawn independently from the uniform distribution on the unit sphere $\mathbb{S}^{D-1}$. For each direction $u_k$, we compute

$$\mathcal{C}_k = \frac{1}{BT}\sum_{i=1}^{BT}((\pi_k^{\mathbf{X},\uparrow})_i - (\pi_k^{\mathbf{Y},\uparrow})_i)^2 \tag{14}$$

where $\pi_k^{\mathbf{X}} = \{\langle \mathbf{X}_{b,t,:}, u_k\rangle\}_{b,t}$ and $\pi_k^{\mathbf{Y}} = \{\langle \mathbf{Y}_{b,t,:}, u_k\rangle\}_{b,t}$ are the collections of projected scalars, and superscript $\uparrow$ denotes sorting.

To establish concentration, we first bound the range of each $\mathcal{C}_k$. By the bounded support assumption, all features after projection satisfy

$$|\langle \mathbf{X}_{b,t,:}, u_k\rangle| \leq \|\mathbf{X}_{b,t,:}\|_2 \cdot \|u_k\|_2 \leq R \cdot 1 = R \tag{15}$$

and similarly for $\mathbf{Y}$. This implies that all projected scalars lie in the interval $[-R, R]$.

The sorting operation preserves this range, so the $i$-th elements $(\pi_k^{\mathbf{X},\uparrow})_i$ and $(\pi_k^{\mathbf{Y},\uparrow})_i$ of the sorted sequences also lie in $[-R, R]$. Consequently, their difference satisfies

$$|(\pi_k^{\mathbf{X},\uparrow})_i - (\pi_k^{\mathbf{Y},\uparrow})_i| \leq |(\pi_k^{\mathbf{X},\uparrow})_i| + |(\pi_k^{\mathbf{Y},\uparrow})_i| \leq 2R \tag{16}$$

Squaring this inequality yields

$$((\pi_k^{\mathbf{X},\uparrow})_i - (\pi_k^{\mathbf{Y},\uparrow})_i)^2 \leq 4R^2 \tag{17}$$

Since $\mathcal{C}_k$ is the average of $BT$ such squared differences, we obtain the bound

$$0 \leq \mathcal{C}_k \leq \frac{1}{BT}\sum_{i=1}^{BT} 4R^2 = 4R^2 \tag{18}$$

The key observation is that, conditional on fixed observation and action features $\mathbf{H}$ and $\mathbf{A}$, all randomness comes from the independently drawn directions $\{u_k\}_{k=1}^K$. Since each direction is drawn

from the same distribution Uniform($\mathbb{S}^{D-1}$), the distance estimates $\{\mathcal{C}_k\}_{k=1}^K$ are independent and identically distributed random variables, each bounded in the interval $[0, 4R^2]$.

We can now apply Hoeffding's inequality, which for bounded i.i.d. random variables $\mathcal{C}_1, \ldots, \mathcal{C}_K \in [0, 4R^2]$ states:

$$\mathbb{P}\left[\left|\frac{1}{K}\sum_{k=1}^K \mathcal{C}_k - \mathbb{E}[\mathcal{C}_k]\right| > \epsilon\right] \leq 2\exp\left(-\frac{2K\epsilon^2}{(b-a)^2}\right) \tag{19}$$

where $[a, b] = [0, 4R^2]$ is the range.

Substituting $b - a = 4R^2 - 0 = 4R^2$:

$$\mathbb{P}\left[\left|\frac{1}{K}\sum_{k=1}^K \mathcal{C}_k - \mathbb{E}[\mathcal{C}_k]\right| > \epsilon\right] \leq 2\exp\left(-\frac{2K\epsilon^2}{(4R^2)^2}\right) \tag{20}$$

$$= 2\exp\left(-\frac{K\epsilon^2}{8R^4}\right) \tag{21}$$

This establishes property (i). $\qquad\square$

### A.3.2 PROPERTY (II): GRADIENT SELECTIVITY

*Proof.* The update $\hat{\mathbf{H}} = \mathbf{H} + \gamma \cdot \mathcal{E}(\mathcal{P}_{\text{obs}}(\mathbf{H}) \cdot g)$ with $g = \exp(-\mathcal{C}/\tau)$ gives:

$$\nabla_{\theta_E}\mathcal{L} = \gamma \cdot \frac{\partial \mathcal{L}}{\partial \hat{\mathbf{H}}} \cdot \frac{\partial \mathcal{E}(\mathcal{P}_{\text{obs}}(\mathbf{H}) \cdot g)}{\partial \theta_E} \tag{22}$$

Since $g$ scales the input, with Lipschitz gradients:

$$\|\nabla_{\theta_E}\mathcal{L}\| \leq \gamma \cdot \left\|\frac{\partial \mathcal{L}}{\partial \hat{\mathbf{H}}}\right\| \cdot L_{\mathcal{E}} \cdot g \cdot \|\mathcal{P}_{\text{obs}}(\mathbf{H})\| \tag{23}$$

Taking expectations: $\|\mathbb{E}[\nabla_{\theta_E}\mathcal{L}]\| \leq \gamma C \cdot \mathbb{E}[g]$ where $C$ combines constants. $\qquad\square$

### A.3.3 PROPERTY (III): UPDATE STABILITY

*Proof.* The enhancement $\hat{\mathbf{H}} = \mathbf{H} + \gamma \cdot \mathcal{E}(\mathbf{X} \cdot g)$ with $\mathbf{X} = \mathcal{P}_{\text{obs}}(\mathbf{H})$ gives:

$$\|\hat{\mathbf{H}} - \mathbf{H}\| = \|\gamma \cdot \mathcal{E}(\mathbf{X} \cdot g)\| \tag{24}$$

$$\leq \gamma \cdot L_{\mathcal{E}} \cdot \|\mathbf{X} \cdot g\| \tag{25}$$

$$\leq \gamma \cdot L_{\mathcal{E}} \cdot g \cdot L_{\mathcal{P}} \cdot \|\mathbf{H}\| \tag{26}$$

where $L_{\mathcal{E}}, L_{\mathcal{P}}$ are Lipschitz constants. $\qquad\square$

## B MORE IMPLEMENTATION DETAILS

### B.1 PSEUDO CODE

Algorithms 1 and 2 show the complete GateFlow procedures for training and inference, respectively. During training (Algorithm 1), we have access to ground truth actions $\mathbf{A}^*$ and compute transport distances against these targets to guide feature enhancement. During inference (Algorithm 2), we perform iterative refinement starting from an initial action estimate $\mathbf{A}^{(0)}$ and progressively improve predictions through $N$ iterations of transport-guided gating.

The computational complexity is dominated by the sorting operations, resulting in $O(KBT\log(BT))$ per iteration for inference, where $K$ is the number of projections (typically 10), $B$ is batch size, and $T$ is sequence length. The space complexity is $O(BTD)$ for storing the projected features. In practice, the random directions can be pre-sampled and reused across batches for additional efficiency.

---

**Algorithm 1** GateFlow Training: Transport-Guided Gating with Ground Truth Actions

---

**Require:** Observation features $\mathbf{H} \in \mathbb{R}^{B \times T \times D}$, Ground truth actions $\mathbf{A}^* \in \mathbb{R}^{B \times L \times D}$
**Require:** Projections $\mathcal{P}_{\text{obs}}, \mathcal{P}_{\text{act}}$, Enhancement $\mathcal{E}$, Parameters $K, \tau, \gamma$
**Ensure:** Transport-guided enhanced features $\hat{\mathbf{H}} \in \mathbb{R}^{B \times T \times D}$

1: $\mathbf{X} \leftarrow \mathcal{P}_{\text{obs}}(\mathbf{H})$      // Project observations to transport space
2: $\mathbf{Y}^* \leftarrow \mathcal{P}_{\text{act}}(\mathbf{A}^*)$      // Project ground truth actions to transport space
3: $\bar{\mathbf{y}}^* \leftarrow \frac{1}{L} \sum_{l=1}^{L} \mathbf{Y}_{:,l,:}^*$      // Mean-pool action sequence
4: $\bar{\mathbf{Y}}_{:,t,:}^* \leftarrow \bar{\mathbf{y}}^*$ for all $t \in \{1, \dots, T\}$      // Broadcast to match observations
5: $\mathcal{W} \leftarrow 0$      // Initialize sliced Wasserstein distance
6: **for** $k = 1$ to $K$ **do**
7:     Sample $\mathbf{u}_k \sim \text{Uniform}(\mathbb{S}^{D-1})$      // Random projection direction
8:     $\pi_k^{\mathbf{H}} \leftarrow \{\langle \mathbf{X}_{b,t,:}, \mathbf{u}_k \rangle\}_{b,t}$      // Project observation features
9:     $\pi_k^{\mathbf{A}} \leftarrow \{\langle \bar{\mathbf{Y}}_{b,t,:}^*, \mathbf{u}_k \rangle\}_{b,t}$      // Project ground truth action features
10:    $\mathbf{h}^{\uparrow} \leftarrow \text{sort}(\pi_k^{\mathbf{H}})$      // Sort projected observations
11:    $\mathbf{a}^{\uparrow} \leftarrow \text{sort}(\pi_k^{\mathbf{A}})$      // Sort projected actions
12:    $\mathcal{W} \leftarrow \mathcal{W} + \frac{1}{KBT} \sum_{i=1}^{BT} |\mathbf{h}_i^{\uparrow} - \mathbf{a}_i^{\uparrow}|^2$      // Accumulate transport cost
13: **end for**
14: $g \leftarrow \exp(-\mathcal{W}/\tau)$      // Transport-guided gate: low distance $\rightarrow$ high gate
15: $\hat{\mathbf{H}} \leftarrow \mathbf{H} + \gamma \cdot \mathcal{E}(\mathbf{X} \odot g)$      // Selective enhancement via gating
16: **return** $\hat{\mathbf{H}}$      // Shortcut-suppressed features for training

---

**Algorithm 2** GateFlow Inference: Iterative Action Refinement with Transport Gating

---

**Require:** Observation features $\mathbf{H} \in \mathbb{R}^{B \times T \times D}$, Initial action estimate $\mathbf{A}^{(0)} \in \mathbb{R}^{B \times L \times D}$
**Require:** Projections $\mathcal{P}_{\text{obs}}, \mathcal{P}_{\text{act}}$, Enhancement $\mathcal{E}$, Parameters $K, \tau, \gamma$, Iterations $N$
**Ensure:** Final action prediction $\mathbf{A}^{(N)} \in \mathbb{R}^{B \times L \times D}$

1: **for** $n = 0$ to $N - 1$ **do**
2:     $\mathbf{X} \leftarrow \mathcal{P}_{\text{obs}}(\mathbf{H})$      // Project observations to transport space
3:     $\mathbf{Y}^{(n)} \leftarrow \mathcal{P}_{\text{act}}(\mathbf{A}^{(n)})$      // Project current action estimate
4:     $\bar{\mathbf{y}}^{(n)} \leftarrow \frac{1}{L} \sum_{l=1}^{L} \mathbf{Y}_{:,l,:}^{(n)}$      // Mean-pool action sequence
5:     $\bar{\mathbf{Y}}_{:,t,:}^{(n)} \leftarrow \bar{\mathbf{y}}^{(n)}$ for all $t \in \{1, \dots, T\}$      // Broadcast to match observations
6:     $\mathcal{W}^{(n)} \leftarrow 0$      // Initialize sliced Wasserstein distance
7:     **for** $k = 1$ to $K$ **do**
8:       Sample $\mathbf{u}_k \sim \text{Uniform}(\mathbb{S}^{D-1})$      // Random projection direction
9:       $\pi_k^{\mathbf{H}} \leftarrow \{\langle \mathbf{X}_{b,t,:}, \mathbf{u}_k \rangle\}_{b,t}$      // Project observation features
10:      $\pi_k^{\mathbf{A}} \leftarrow \{\langle \bar{\mathbf{Y}}_{b,t,:}^{(n)}, \mathbf{u}_k \rangle\}_{b,t}$      // Project action features
11:      $\mathbf{h}^{\uparrow} \leftarrow \text{sort}(\pi_k^{\mathbf{H}})$      // Sort projected observations
12:      $\mathbf{a}^{\uparrow} \leftarrow \text{sort}(\pi_k^{\mathbf{A}})$      // Sort projected actions
13:      $\mathcal{W}^{(n)} \leftarrow \mathcal{W}^{(n)} + \frac{1}{KBT} \sum_{i=1}^{BT} |\mathbf{h}_i^{\uparrow} - \mathbf{a}_i^{\uparrow}|^2$      // Accumulate transport cost
14:     **end for**
15:     $g^{(n)} \leftarrow \exp(-\mathcal{W}^{(n)}/\tau)$      // Compute transport-guided gate
16:     $\hat{\mathbf{H}}^{(n)} \leftarrow \mathbf{H} + \gamma \cdot \mathcal{E}(\mathbf{X} \odot g^{(n)})$      // Enhance features via gating
17:     $\mathbf{A}^{(n+1)} \leftarrow \text{FlowModel}(\hat{\mathbf{H}}^{(n)})$      // Predict refined actions
18: **end for**
19: **return** $\mathbf{A}^{(N)}$      // Final action prediction after refinement

---

## B.2 Evaluation Details

**LIBERO Benchmark Details:** The LIBERO benchmark consists of four distinct task suites, each designed to evaluate different aspects of model robustness and generalization capabilities:

**LIBERO-Spatial** contains 10 tasks testing spatial reasoning with varied object placements and orientations, requiring models to generalize across different spatial configurations while maintaining

task success. Tasks include placing objects at specific locations, arranging items in patterns, and manipulating objects with precise spatial requirements.

**LIBERO-Object** evaluates object-centric reasoning through 10 tasks involving novel shapes, textures, and physical properties, challenging models to handle unseen object variations. This suite tests whether models can adapt to new visual appearances while maintaining functional understanding of manipulation primitives.

**LIBERO-Goal** assesses goal-oriented execution through 10 tasks with different task objectives and success criteria, testing compositional understanding of natural language instructions. Tasks require parsing complex instructions and executing appropriate action sequences to achieve specified goals.

**LIBERO-Long** contains 10 tasks requiring extended multi-stage planning with sequential subgoals, particularly challenging for models that rely on shortcuts rather than true task understanding. These long-horizon tasks expose failure modes where models memorize action sequences instead of developing genuine planning capabilities.

## B.3 HYPERPARAMETERS

This section provides comprehensive hyperparameter details for reproducing GF-VLA training and evaluation. Our configuration is based on the Pi0.5 architecture with GateFlow enhancements.

Table 4: **Complete GF-VLA Hyperparameter Configuration.** All hyperparameters used for training and evaluation on LIBERO benchmarks. Values marked with * indicate hyperparameters that were tuned based on sensitivity analysis.

| Category | Parameter | Value |
|---|---|---|
| **Training Configuration** | Global batch size | 256 |
| | Number of training steps | 30,000 |
| | Number of data workers | 2 |
| | Training precision | bfloat16 |
| | FSDP devices | 8 (A100 GPUs) |
| **Optimizer** | Optimizer type | AdamW |
| | Peak learning rate | 5e-5 |
| | Learning rate schedule | Cosine decay |
| | Warmup steps | 10,000 |
| | Gradient clip norm | 1.0 |
| | EMA decay | 0.999 |
| **Model Architecture** | Base model | Pi0.5 (Pi0Config with pi05=True) |
| | Action horizon | 10 |
| | Action dimension | 7 (LIBERO) |
| | Discrete state input | False |
| **GateFlow Parameters** | Number of sliced projections ($K$)* | 32 (range: 4-32) |
| | Enhancement strength ($\gamma$)* | 0.4 (range: 0.1-1.0) |
| | Temperature ($\tau$) | 1.0 |
| | Refinement iterations | 2 |
| | Refinement warmup steps | 1,000 |

## C  BROADER IMPACT

This work contributes to advancing Vision-Language-Action models for robotics by addressing the fundamental problem of shortcut learning. The proposed GateFlow method has several broader implications.

Our approach enables more robust and generalizable robotic systems by mitigating spurious correlations that cause catastrophic failures in real-world deployment. This could accelerate the development of reliable household and industrial robots, potentially improving quality of life and productivity. The method's compatibility with existing VLA architectures allows for easier adoption across the robotics community.

As with any advancement in robotic capabilities, improved VLA models may contribute to automation that could displace certain jobs, requiring careful consideration of societal impacts. Additionally, more capable robotic systems necessitate robust safety measures and ethical guidelines to prevent misuse. While our method adds modest computational overhead, the improved generalization may reduce the need for extensive retraining on new tasks or environments, potentially reducing overall computational costs in the long term.

## D  ETHICS STATEMENT

This research adheres to standard ethical guidelines for machine learning research. Our work does not involve human subjects, animal experimentation, or collection of personal data. All experiments are conducted using publicly available datasets (LIBERO) and standard benchmarks. The research aims to improve the reliability and safety of robotic systems, which aligns with beneficial applications of AI.

We acknowledge the computational resources required for our experiments and have made efforts to optimize efficiency. All code and experimental details will be made available to facilitate reproducibility and further research in the community.

## E  USE OF LARGE LANGUAGE MODELS

In this paper, large language models (LLMs) are employed solely for writing assistance purposes. Specifically, LLMs are used to polish the text, improve clarity of expression, and refine the presentation of ideas. They are not involved in the design of methods, implementation of experiments, analysis of results, or any other part of the scientific contributions.

