# OpenReview forum: "GateFlow: Mitigating Shortcut Learning in VLA Models via Gated Flow Matching"
_ICLR.cc/2026/Conference — ICLR 2026 Conference Withdrawn Submission_

### Official Review · Reviewer_6uQe · 2025-10-31

**Soundness:** 1
**Presentation:** 2
**Contribution:** 2
**Rating:** 4
**Confidence:** 4

**Summary:**

This paper proposes GATEFLOW for Vision-Language-Action (VLA) models. GATEFLOW introduces the use of the sliced Wasserstein distance to measure the discrepancy between the VLM-generated observation feature $H$ and the ground-truth action. This distance is then used as a signal to guide the VLM’s understanding and refine $H$, aiming to improve model performance. The authors claim that this approach mitigates shortcut learning in VLA models and enhances their generalization ability.

**Strengths:**

1. Although I am not an expert in the VLA domain, to the best of my knowledge, the proposed method appears novel and clearly distinct from prior approaches such as OT-CFM.

2. The idea of leveraging a Wasserstein-based metric to control representation alignment is interesting and may inspire further exploration in multimodal learning.

**Weaknesses:**

The motivation of the paper is not fully convincing. My main concerns are as follows:

(a) The paper builds on the practical framework of flow matching but heavily emphasizes the ELBO–NLL gap in its theoretical motivation. However, the FM loss is not directly equivalent to the ELBO derived for minimizing the NLL. The authors seem to rely on the latent equivalence between diffusion models and flow matching, as well as the ELBO formulation in diffusion modeling, to construct their argument. Without rigorous mathematical derivations (or at least clear references) this logical connection remains unsupported.

(b) The paper attributes the lack of generalization (or the presence of shortcut learning, which I view as a form of memorization) in VLA models directly to the ELBO–NLL gap. However, this claim is not intuitively convincing (L190). The ELBO loss generally leads to good generalization in most diffusion-based generative models. This raises the question of whether the poor generalization in VLA actually stems from data sparsity rather than the ELBO–NLL gap. Prior work [1] has shown that diffusion-based models tend to memorize under insufficient training data, which may better explain the observed shortcut issue.

(c) On p.4, L208, the authors state that “when features are semantically aligned through genuine understanding, this transformation is smooth and requires little effort, resulting in low W.” This statement lacks sufficient theoretical or empirical justification. From my perspective,  $H$ acts as a guidance signal for the flow model, while $A$ is the flow model’s action output. It is unclear why requiring $H$ and $A$ to be close in the Wasserstein sense would necessarily reduce shortcut learning or memorization. Could the authors provide additional experiments or analyses to clarify this connection?

(d) In the Appendix, the proof of Theorem 4.1(particularly Property 3 (Convergence to NLL-optimal solutions) and proof of Property 3 (Update Stability)) is quite confusing. It is unclear how Eq. (26) directly implies that the ELBO–NLL gap decreases monotonically. Moreover, Property 1 merely establishes that the sliced Wasserstein distance approximates the true Wasserstein distance, which makes the claim in L308 (“sliced Wasserstein approximation reliably distinguishes shortcuts from semantic features”) questionable.

(e) Regarding the experiments, in Table 1, GATEFLOW shows significant improvement only on long-horizon planning tasks (noting that the authors build on the second-best model $\pi\_{0.5}$-base). Hence, the gains over earlier methods may not be solely attributed to GATEFLOW itself. On the other three metrics, the results are roughly comparable to the baselines, which further raises doubts about whether GATEFLOW truly mitigates shortcut learning.

[1] The Emergence of Reproducibility and Generalizability in Diffusion Models

**Questions:**

1. Can the authors provide an ablation or correlation analysis demonstrating that the shortcut phenomenon is positively correlated with W(H, A)? For example, provide W(H, A) in Table 3.

2. Have the authors considered alternative approximations to the Wasserstein distance, such as the Sinkhorn divergence? Theoretically, a more accurate approximation could lead to a more precise gating signal.

---

### Official Review · Reviewer_hV2h · 2025-10-31

**Soundness:** 1
**Presentation:** 4
**Contribution:** 3
**Rating:** 4
**Confidence:** 3

**Summary:**

The paper argues that, while minimizing an ELBO wrt to the true likelihood, VLA policies trained with flow matching (or diffusion) end up finding "shortcuts", due to spurious correlations. The authors propose a method, GateFlow, based on optimal transport, to circumvent this issue. GateFlow computes the Wasserstein distance between observation features and ground truth actions to identify semantic alignment. This distance controls a gate that selectively modulates flow matching from noise to predicted action. The method is evaluated on the LIBERO benchmark.

**Strengths:**

* **Originality**: I think the idea is original and analyzes one aspect of VLAs that is generally overlooked. This is the main strength of the paper.
* **Theoretically-grounded**: the authors provide a theoretical analysis of their methods, guaranteeing desirable properties, e.g. that GateFlow reduces the ELBO-NLL gap.

**Weaknesses:**

* **Motivation**: while intuitively the idea proposed makes sense, there's no empirical evidence that the issue with VLAs is that they are minimizing the "shortcut distance" in the ELBO. The proposed method demonstrating higher performance in perturbed settings does not necessarily imply that, nor there is any analysis that justifies it.
* **Evaluation**: the method is implemented on top of pi0.5 and evaluated on LIBERO, where performance is saturated (~97% before applying GateFlow). First, I would argue that it would be a more sensible choice to apply GateFlow on top of a VLM, rather on top of a VLA, already fine-tuned with an action prediction objective, as the latter might have already lost semantic information in the visual features. Secondly, I think that evaluating with other VLMs or environments would strongly strengthen the experimental section.

**Questions:**

* Does the approach only work with flow-matching/diffusion-based policies? In theory, the gating mechanism should work for other prediction heads as well?
* How do the authors compute/approximate sliced-Wassertein distances?

I enjoyed the idea presented in the paper and reading the paper itself, and I would be happy to increase my score, if my concerns were addressed during rebuttal.

---

### Official Review · Reviewer_Renq · 2025-11-01

**Soundness:** 1
**Presentation:** 2
**Contribution:** 2
**Rating:** 2
**Confidence:** 4

**Summary:**

This paper proposes GateFlow, a transport-guided gating mechanism to mitigate shortcut learning in Vision-Language-Action (VLA) models. The authors argue that the optimization gap between ELBO and true NLL in flow-matching-based VLA training encourages spurious correlations between visual patterns and actions. GateFlow measures the Wasserstein distance between observation and action features to identify and suppress shortcut pathways, selectively amplifying features corresponding to genuine semantic understanding. The paper provides theoretical analysis and experiments on LIBERO benchmarks, reporting improved performance.

**Strengths:**

1. The paper highlights an important limitation of current VLA models: their tendency to overfit to spurious correlations (shortcut learning). The link between the ELBO–NLL gap and shortcut behavior is conceptually clear and well-motivated.
2. Employing the Wasserstein distance as a measure of semantic alignment between different features is an interesting idea.
3. The GateFlow module can be plugged into existing VLA architectures.

**Weaknesses:**

1. The paper’s statement about mitigating “shortcut learning” to enhance semantic understanding needs stronger argument. For VLAs with System1 and System2 structures, semantic reasoning primarily happens in a symbolic or high-level semantic layer (system 2). It is unclear whether deeper reasoning are needed for system1. Furthermore,  in practice, large-scale and diverse datasets, along with robust pretrained VLM features, already mitigate shortcuts substantially. It’s unclear why this architectural modification is necessary instead of improving data diversity or leveraging VLM pretraining better. A discussion comparing these two directions (data scaling vs. gating) is missing.
2. The proposed alignment between observation features and actions is conceptually questionable.
* There can be multiple valid actions for the same observation. Thus, their alignment may not represent true semantics. A more meaningful alignment might be between vision–language embeddings and actions, as that reflects task-level semantics. Also, since actions are per timestep, semantic correspondence is inherently weak.
* The action features are not pretrained rather learned. Consider an experiment where you always put a green object for one specific task and no green object for the others, then when you train the model with such data, the action representation will be aligned with the observation feature during the training process. At inference time, when the green object is present, the model can just ignore the language instruction and perform the specific task. I would like to see such controlled experiments to validate the proposed method can suppress such shortcuts.
3. It’s unclear what exactly constitutes “observation features”. Are they derived purely from the visual encoder, or do they include language and proprioceptive states? Similarly, “action features” are said to be used in the Wasserstein computation, but since actions are outputs, it is unclear how their features are extracted before prediction. Also, it’s unclear whether gating is applied per transformer layer.
4. The overall performance gain over the strong π0.5 baseline is modest (e.g., +1.4% average, +4% on LIBERO-Long). Given that the method adds additional computations and hyperparameters, the improvement seems incremental rather than transformative.
5. The use of LIBERO as the main benchmark is not ideal for demonstrating shortcut mitigation. Not only because model performance on this benchmark is saturated, but also because the simulated scenes are well controlled and less prone to the vision-action shortcut correlations. Experiments with real-world experiments with lighting changes or texture variations and simulation experiments with a dataset includes a spurious correlation (e.g., the experiment of a green object always associated with a specific task as aforementioned ) should be included to validate the method.
6. Missing related works[1,2,3] on mitigating the shortcut.

[1] Xing, Youguang, et al. "Shortcut learning in generalist robot policies: The role of dataset diversity and fragmentation." arXiv preprint arXiv:2508.06426 (2025).

[2] Lin, Fanqi, et al. "OneTwoVLA: A Unified Vision-Language-Action Model with Adaptive Reasoning." arXiv preprint arXiv:2505.11917 (2025).

[3] Huang, Huang, et al. "Otter: A vision-language-action model with text-aware visual feature extraction." arXiv preprint arXiv:2503.03734 (2025).

**Questions:**

see weakness 3, the definition of observation features and action features, and the calculation process is unclear.

---

### Official Review · Reviewer_d5vr · 2025-11-03

**Soundness:** 3
**Presentation:** 3
**Contribution:** 2
**Rating:** 6
**Confidence:** 3

**Summary:**

This paper tackles the shortcut learning problem in vision-language-action (VLA) models, where models rely on spurious correlations instead of true task understanding. The authors trace this issue to the gap between the ELBO and the true negative log-likelihood (NLL) in flow-matching based training, which allows models to minimize the training loss through memorization rather than genuine reasoning, leading to fragile and non-generalizable behaviors.

To address this, authors propose GateFlow, a transport-guided gating mechanism that measures the Wasserstein distance between observation and action representations to detect shortcuts. Low transport distance indicates semantic alignment, while high distance reveals spurious correlations. GateFlow converts this distance into gating weights that enhance meaningful features and suppress shortcuts, improving model understanding. It also integrates easily into existing VLA models and uses ground-truth actions during training, then applies iterative refinement at inference to maintain alignment.

Authors conduct comprehensive experiments on LIBERO, a standardized benchmark for evaluating robotic manipulation skills with a focus on generalization. Compared to existing baselines, GateFlow shows particularly large gains in long-horizon planning tasks where shortcut-induced errors tend to accumulate.

**Strengths:**

1. The paper provides analysis of shortcut learning in VLA models, identifying it as a byproduct of optimizing the ELBO rather than the true NLL in flow-matching frameworks. It introduces GateFlow, a transport-guided gating mechanism based on Wasserstein distance, represents a principled solution that bridges geometric understanding with causal feature alignment. Different from previous works, this formulation offers a new lens to interpret and mitigate spurious correlations in multimodal policy learning.
2. The paper is clearly written and well-structured. The methodology part includes both empirical insights such as implementation details and theoretical justifications. The experiment part consists of comprehensive comparisons to baselines and systematic ablation studies.
3. Experimental results are convincing. GateFlow shows strong overall performance and clear robustness across diverse tasks on the LIBERO benchmark. Its largest improvements on long-horizon tasks demonstrate that GateFlow effectively mitigates shortcut-induced errors and enhances true sequential reasoning.

**Weaknesses:**

1. Lack of real robot experiments. Since this approach focuses on improving VLA models, it would be beneficial to see real world experimental results.
2. Generality of approach. It seems that GateFlow is specifically designed and combined with VLA models using flow-based action predictor. Is there any possibility to apply similar ideas to other action head architectures? Such as autoregressive models or diffusion-based models?

**Questions:**

1. I wonder what are intuitions behind iterative refinement procedures applied in inference time? On one hand, whether using predicted action from previous timestep $\hat{a}_{t-1}$ will introduce additional noises or errors; on the other hand, since this is an iterative approach, whether this will increase inference time cost. I am not very convinced about the design choice here.
2. Since the goal of using Wasserstein distance is to compute semantic similarity, will other distance metric be applicable as well? What are advantages of Wasserstein distance compared to other metrics under this setting?

---

### Note · Authors · 2025-11-14

I have read and agree with the venue's withdrawal policy on behalf of myself and my co-authors.